# Blood Sugar, Haemoglobin and Malondialdehyde Levels in Diabetic White Rats Fed a Diet of Corn Flour Cookies

**DOI:** 10.3390/foods11121819

**Published:** 2022-06-20

**Authors:** Nur Aini, Budi Sustriawan, Nadia Wahyuningsih, Ervina Mela

**Affiliations:** Department of Food Science and Technology, Jenderal Soedirman University, Purwokerto 53123, Indonesia; budi.sustriawan@unsoed.ac.id (B.S.); nadia.wahyuningsih@mhs.unsoed.ac.id (N.W.); ervina.mela@unsoed.ac.id (E.M.)

**Keywords:** blood sugar, coconut sugar, corn biscuit, diabetic, gluten-free, haemoglobin, malondialdehyde, virgin coconut oil

## Abstract

The purpose of the study was to analyse the chemical composition of corn cookies containing different types of sugar and fat, and determine their effect on physiological parameters in diabetic rats. The experimental animals were studied using a randomised block design with seven groups of rats. The test groups were as follows: group 1, negative control rats (normal) fed standard; group 2, positive control rats (diabetic) fed standard; group 3, diabetic rats fed wheat cookies; group 4, diabetic rats fed C1 corn cookies; group 5, diabetic rats fed C2 corn cookies; group 6, diabetic rats fed C3 corn cookies; and group 7, diabetic rats fed C4 corn cookies. The tests on the rats revealed that the cookies had significant effects on blood sugar, malondialdehyde (MDA) and haemoglobin levels as well as body weight parameters. Corn cookies containing crystalline coconut sugar and virgin coconut oil (VCO) were effective at lowering blood sugar and MDA levels while increasing haemoglobin and body weight in diabetic rats. Significantly, after four weeks on this diet, rats with diabetes mellitus were in the same overall condition as normal rats. These findings suggest that these cookies may be gluten-free functional foods suitable for diabetics. These findings suggest that diabetics can safely consume maize cookies.

## 1. Introduction

The prevalence of diabetes mellitus (DM) is 1.4% in 20- to 24-year-olds, 19.9% in 75- to 79-year-olds and 19.3% in 65- to 99-year-olds, representing 135.6 million cases in this age group alone [1]. If current trends continue, by 2030 diabetes will affect 20.4% and 20.5% of people aged 20–24 and 75–79 years, respectively, while numbers of over-65s with diabetes are projected to reach 195.2 million by 2030 and 276.2 million by 2045. Indonesia is ranked in the top ten countries in the world for the number of diabetics, trailing only European countries, the US, and China [2].

Type 2 diabetes affects 90 percent of diabetic patients, and 53 percent of diabetics are unaware that they have the disease [3]. Type 2 diabetes is caused by the body’s insulin failing to function properly. Unhealthy lifestyles, such as poor food balance, lack of physical activity, stress, and hereditary diseases, can all contribute to the onset of type 2 diabetes. Meal planning, exercise, and weight loss are the initial steps in controlling type 2 diabetes [4]. Strategies to regulate diet and control blood glucose include the consumption of foods that do not cause a rapid increase in blood glucose [5]. Eating arrangements for diabetic patients can be made using a glycaemic index (GI) approach [6]. As a result, food products made from low-GI raw materials must be modified so that they can still contribute to nutritional adequacy for people with diabetes. Cookies are enjoyed by people of all ages [7], and may be considered to be a functional food if they have health-promoting properties such as low GI and the ability to assist in controlling blood glucose levels. These functional properties can be achieved by making changes to the main ingredient, namely replacing wheat flour with other ingredients to create gluten-free cookies with higher fibre content and low GI [8,9].

Corn is a gluten-free cereal that can regulate blood sugar levels; therefore, processed corn products are expected to lower blood sugar levels. On the other hand, effective control of blood sugar levels in diabetic Wistar rats fed a diet of corn flour with tempeh flour supplementation was demonstrated [10]. Corn flour has a low GI (48) and total sugar (2.98%), so it does not raise blood sugar levels [11]. According to [12], foods with a GI of less than 50 are included in the low GI group.

Corn flour can be used as the primary raw material in the production of cookies. The main benefit of corn flour as a food ingredient is its higher level of dietary fibre in comparison with wheat flour [13], which is beneficial for diabetics. As a result, cookies made with corn flour are expected to have a low GI value and thus aid in the control of blood glucose.

Sugar is used in the production of cookies, and in addition to serving as a sweetener, it also plays a role in determining the spread of the fracture structure. Several sweeteners can be used in the production of cookies, including coconut, palm, and granulated sugars [14]. Different types of sugar have differing effects on the colour of both outer and inner sections of cookies, as well as on flavour, aroma, and texture [15]. Coconut sugar has a lower GI, which is 35, and a calorie content of 368 kcal, compared with granulated sugar (GI of 58 and calorie content of 400; [16]). The sugar content of crystalline coconut sugar is also lower than that of granulated sugar, making it ideal for diabetics and capable of lowering unsaturated fat levels in the body [17,18].

Fat (commonly margarine) is added to achieve a soft texture in cookies [19,20]. Virgin coconut oil (VCO) can also be used as a fat source, with a texture and flavour similar to that of margarine [21]. The use of VCO has certain advantages, e.g., it contains high levels of medium-chain triglycerides (MCTs), which are beneficial for diabetics needing to limit the amount of saturated fat in their diet. The MCTs found in VCO have been shown to gradually regenerate pancreatic beta cells, stimulating insulin production and improving insulin sensitivity [22].

One of the causes of type 2 diabetes is oxidative stress, which can induce insulin resistance in peripheral tissues and impair insulin secretion from pancreatic beta cells. Oxidative stress will cause lipid peroxidation of cell membranes. The lipid peroxidation of cell membranes will make it easier for erythrocytes to undergo haemolyses, resulting in free haemoglobin and hence lowering haemoglobin levels. Hyperglycaemia produces an increase in free radicals in cells, which can lead to oxidative stress and the formation of Reactive Oxygen Species (ROS) or Reactive Nitrogen Species (RNS). This oxidative stress hastens the onset and progression of diabetes. Malondialdehyde levels will rise as a result of the increase in free radicals in the cell membrane. Therefore, food for people with DM2 is also expected to prevent a decrease in haemoglobin and increase malondialdehyde.

Until now, no in vivo research has been conducted to investigate the effect of non-gluten cookies made with different types of sugar (sugar and coconut sugar) or fat (margarine and VCO) on levels of blood sugar, malondialdehyde (MDA), or haemoglobin and body weight in experimental animals. In vivo research is required before the effect of such cookies can be determined in humans.

The purpose of this study was to determine the chemical composition of corn cookies containing different types of sugar and fat and examine their effect on blood sugar levels, MDA, haemoglobin, and body weight in diabetic rats.

## 2. Materials and Methods

### 2.1. Materials

Corn flour, wheat flour (Kunci Biru brand, PT Indofood Sukses Makmur Tbk.), sugarcane, crystalline coconut sugar, baking powder, eggs, margarine, VCO (PT Mutia), milk powder and salt were used to make the cookies. Wistar rats, standard feed, and other analytical materials were among the materials used in the analysis, while tools for cookie creation and analysis were among those employed.

### 2.2. Production of Cookies

Cookies were produced according to the formula described by [21] with modifications. Five types of cookies were created: corn cookies made with granulated sugar and margarine (C1), corn cookies made with crystalline coconut sugar and margarine (C2), corn cookies made with granulated sugar and VCO (C3), corn cookies made with crystalline coconut sugar and VCO (C4), and wheat flour cookies made with granulated sugar and margarine as a control (W).

### 2.3. Analysis of Nutritional Value

Iron content was determined using the spectrophotometric method [23], dietary fibre content was determined using the enzymatic method [24], beta-carotene content was determined using a spectrophotometer [25], total sugar content was determined using the anthrone method [26], and proximate levels (water, ash, fat, protein and carbohydrates) were determined using the AOAC method [27].

### 2.4. Analysis of Experimental Animals

Analysis of experimental animals, namely Wistar rats, was carried out in three stages: test feed preparation, experimental animal preparation, and testing. The experimental animals were prepared by adapting 28 male Wistar rats (aged 2 months), and housing them individually for three days. During the adaptation period, the rats were given AD II standard feed and allowed to drink ad libitum. Body weight was recorded on the last day of the adaptation period as initial data prior to the induction period.

The rats were then treated with streptozotocin at a dose of 45 mg/kg body weight, nicotinamide at a dose of 110 mg/kg and citrate buffer at a dose of 3 mL/200 g body weight (injected intraperitoneally) to raise blood sugar levels to 200 mg/dL. The induction period lasted three days. At the end of the third day, blood sugar levels and body weight were measured as baseline data. The rats were allowed standard feed ad libitum during the induction period.

The rats were divided into seven groups of four, with test groups as follows: group 1, negative/normal control rats given standard AD II feed; group 2, positive/diabetic control rats given standard AD II feed; group 3, diabetic rats fed wheat cookies (W); group 4, diabetic rats fed C1 corn cookies; group 5, diabetic rats fed C2 corn cookies; group 6, diabetic rats fed C3 corn cookies; and group 7, diabetic rats fed C4 corn cookies. Ad libitum feeding continued for four weeks (28 days).

Body weight was measured using the technique outlined by [28], while blood sugar was measured using the enzymatic glucose oxidase-phenol amino phenazone (GOD-PAP) method [29], and MDA levels were measured using a spectrophotometric assay [30]. Levels of haemoglobin were also measured using a spectrophotometer [31].

For the GOD-PAP method (measurement of blood sugar levels), 0.5 mL of blood was taken from each test animal through the eye canthus (under the eyeball) and placed in an Eppendorf tube. After centrifugation of blood samples at 210× *g* for 10 min, 0.01 mL of supernatant was transferred to a test tube; 1 mL of GOD-FS reagent was then added, fortified, and allowed to stand for 10 min. Absorbance was measured by spectrophotometry at a wavelength of 500 nm.

For spectrophotometric analysis of haemoglobin levels, intracardiac blood (1 mL) was drawn on day 16 (24 h after induction) and placed in an EDTA-containing tube; haemoglobin was analysed immediately. The oxyhaemoglobin method was used to calculate concentration: 5 mL of 0.1% sodium carbonate solution was added to the blood in a test tube, followed by 20 L of EDTA; the test tube was then sealed and shaken for 10 s, and absorption was measured at 540 nm. Haemoglobin concentration measurements were carried out using previously prepared calibration cuvettes.

The thiobarbituric acid (TBA) reactive substance method was used to measure MDA levels. The principle of this method is based on the ability of MDA and TBA to form a pink complex. Using a Teflon Potter-Elvehjem homogeniser, 1 g of liver tissue was homogenised with 9.0 mL of 1.15% KCl solution. Then, 0.2 mL of liver homogenate, 0.2 mL of 8.1% sodium dodecyl sulfate, 1.5 mL of 20% acetic acid solution (adjusted to pH 3.5) and 1.5 mL of 0.8% TBA solution were mixed with 4.0 mL of water, heated at 100 °C for 60 min and cooled in an ice bath. After cooling, the mixture was shaken with 1.0 mL of water and 5.0 mL of n-butanol:pyridine (15:1 *v*/*v*). After centrifugation of the mixture at 4000 rpm for 10 min, the organic layer was removed and absorbance was measured at 532 nm using a visible spectrophotometer.

### 2.5. Research Design and Statistical Analysis

The study used experimental methods and testing the chemical properties of cookies using a single Completely Randomized Design (CRD) with 5 levels, namely corn cookies made with granulated sugar and margarine (C1), corn cookies made with crystalline coconut sugar and margarine (C2), corn cookies made with granulated sugar and VCO (C3), corn cookies made with crystalline coconut sugar and VCO (C4), and wheat flour cookies made with granulated sugar and margarine as a control (W). Each was repeated 3 times to obtain 15 experimental units. Research on experimental animals used a randomized block design with 7 groups of mice. Each group was repeated 5 times so that 35 Wistar rats were needed.

The research data were analyzed using analysis of variance (ANOVA) at the 95% confidence level and if there was a significant effect, then continued Duncan Multiple Range Test (DMRT) with 95% level.

### 2.6. Ethical Considerations

Animal management was carried out in accordance with the European Communities Council Directive 86/609/EEC guidelines for the care and use of experimental animals and the official Mexican Standard (NOM-062-ZOO-100-1999) technical specifications for the production, care, and use of laboratory animals (Diario Oficial de la Federación, 2001). Similarly, the Ethics Commission of the Faculty of Medicine, Jenderal Soedirman University (No. 066/KEPK/I/2022), approved the project. Individual animals were housed under controlled temperature conditions (24 °C), light–dark cycles (12/12 h), and ad libitum food and water.

## 3. Results and Discussion

### 3.1. Nutritional Composition

Moisture content was higher in the corn flour cookies C1 and C2 (3.9% and 4.4%, respectively) than in the control cookies (W) made with wheat (3.7%; see Table 1). This was due to the fact that corn flour absorbs more water than wheat flour. According to Liu et al. [32] water absorption is influenced by amylose, which has a smaller molecular size than amylopectin, thus facilitating water absorption [33]. This is consistent with the findings of Wesley et al. [34] who showed that biscuits made with a higher proportion of corn flour absorbed more water than those made with wheat flour due to the higher amylose content of corn flour.

Due to the higher water content of margarine (1.5%) compared with VCO (0.22–0.36%), cookies made with VCO (specifically C3 and C4) contained less water than those made with margarine [35]. These results are consistent with those of Nurani and Yuwono [36], who stated that the greater the amount of margarine added to taro cookies, the higher the water content. According to Sustriawan et al. [20] the difference in water content between margarine and butter accounts for the water content of bread with the addition of 16% VCO.

In addition, cookies made with granulated sugar (C1 and C3) contained less water than cookies made with crystalline coconut sugar (C2 and C4). This was because of the hygroscopic properties of crystalline coconut sugar, which easily attracts water (causing it to become “mushy”); the presence of polyhydroxy groups, which form hydrogen bonds with water, contributes to hygroscopicity [17].

In general, ash content was higher for corn cookies (1.5–1.9%) than for wheat cookies (1.3%, see Table 1) due to the higher ash content of corn flour compared with wheat flour (0.79–1.01% vs. 0.59%; [8]. This is similar to the findings of [21], who showed that the higher ash content of cookies made with 100% corn flour (2.12%), compared with cookies made with wheat flour (0.98%), was due to higher levels of essential minerals Fe, Ca, and P. In addition, Dewi et al. [14] demonstrated higher ash content for cookies made with 80% corn flour and 20% black rice bran, in comparison with cookies made with 100% wheat flour. Again, this was due to the higher ash content of black rice bran and corn flour (6.6–9.9% and 1.8%, respectively), relative to wheat flour (0.52%).

In addition, corn cookies made with crystalline coconut sugar and margarine (C2) had a higher ash content than wheat cookies and other corn cookies as a result of the higher ash content of crystalline coconut sugar (0.92–2.58%) compared with granulated sugar (0.10–1.15%) (see Table 1). Similarly, [21] showed that sorghum flour cookies made with crystalline coconut sugar had a higher ash content (1.83%) than cookies made with granulated sugar (1.40%).

The use of margarine, which is mineralised during the manufacturing process, prevents excessive heat. Sustriawan et al. [20] showed that cookies made with margarine had higher ash content (0.67%) than cookies made using a mixture of margarine and VCO (0.40%) or cookies made with VCO alone (0.26%). This is also supported by the results of [37], who showed that biscuits made with margarine had a higher absorption rate of 1.46–1.62%, compared with biscuits made without margarine, which had a lower absorption rate of 1.23–1.41%.

In general, the corn flour cookies had a higher fat content than the wheat cookies, with the exception of corn cookies made with granulated sugar and margarine (C1), which had almost the same fat content as wheat cookies (see Table 1). Corn flour has been shown to have higher levels of fat than wheat flour (5.42% vs. 2.09%; [38]. This is consistent with the study of Xie et al. [30] showing that cookies made with 80% corn flour and 20% bran flour had a higher fat content (16.7%) than cookies made with 100% wheat flour (11.2%).

Overall, cookies containing VCO (C3 and C4) had the highest fat content (17.3% and 17.8%, respectively), as the fat content of VCO is 100% (due to water content below 1%); in contrast, margarine is only 81% fat [21]. In addition, the fat content of sorghum cookies made with crystalline sugar has been shown to be higher than that of cookies made with granulated sugar, as crystalline coconut sugar contains 10% fat, while granulated sugar lacks fat altogether [15].

The results of this study are in line with the findings of Sustriawan et al. [20], who showed that sorghum cookies containing higher levels of VCO also had higher fat levels. They discovered that biscuits made with 100% VCO contain more fat than biscuits made with margarine. Although the fat content of biscuits formulated with VCO is relatively high, the fatty acid composition of VCO is beneficial, as 50% of the fatty acids are MCTs (e.g., lauric acid in particular), which are easily absorbed by the body [22].

Regarding protein, the protein content of the corn cookies (19.3–23.7%) was lower than that of the wheat cookies (29.2%) due to the lower overall protein content of corn flour (11.02%), compared with wheat flour (14.45%); these findings are consistent with those of Aini et al. [21] and Hand and Lee [39].

Our results also showed higher levels of fibre in the corn cookies (4.2–6.2%) than the wheat cookies (2%) (see Table 1). This is consistent with a crude fibre content of 4.24% in cornflour and 1.92% in wheat [40]. Again, the findings of this study are in agreement with those of Aini et al. [21], who showed crude fibre levels of 3.67% and 0.98% in cookies made from corn flour and wheat flour, respectively [8].

Foods with levels of high fibre are typically low in calories as well as low in sugar and fat, which can help reduce the occurrence of obesity and heart disease [41]. According to Ahn et al. [42], fibre increases the density and thickness of food in the digestive tract and inhibits the movement of enzymes, slowing the digestive process and lowering the sugar response in people with DM. Dietary fibre also aids in the digestive process, slows glucose absorption, lowers the levels of cholesterol and low-density lipoprotein (LDL) and stimulates the production of short-chain fatty acids.

The sugar content of the wheat cookies (85.4%) was higher than that of the corn flour cookies (63.9–80.4%) as a result of the lower sugar content of corn flour (1.20%; [40]), in comparison with wheat flour (2–3%) [3]. The lowest sugar content was found in corn cookies made using crystalline coconut sugar and VCO (C4), as the total sugar content of coconut sap (the raw material for crystalline coconut sugar) is only 9.30% [17]. By contrast, the raw material for granulated sugar, sugarcane juice, contains 32.42% sucrose, 2.41% fructose, 1.58% glucose, and 2.00% galactose [16]. Thus, lower sugar content was found in corn cookies made with crystalline coconut sugar.

In terms of iron content, higher levels were found in the corn cookies (3.7–4.7%) than in the wheat cookies (2.5%) (see Table 1). This was due to the higher iron content of corn flour (2.4 mg per 100 g; [43]) compared with wheat flour (1.2 mg per 100 g; [44]). The most iron-rich cookie formula (4.7%) was C4, i.e., corn cookies made with coconut sugar crystals and VCO. Apart from the corn flour, the coconut sugar (iron content 1.2 mg/g) and VCO (iron content 1.22–5.91 mg/L) also both contributed to overall iron levels; the iron content of margarine is lower (0.06 mg/100 g) than that of VCO.

As shown in Table 1, the corn flour cookies had higher levels of beta-carotene (1250.4–1877.9 ppm) than the wheat flour cookies (782.3 ppm), in agreement with the higher levels of beta-carotene found in yellow corn flour, compared with wheat flour (510 ppm vs. 9 ppm; [45]. Our findings are consistent with those of [46], who showed the higher the percentage of corn flour used for baking cookies, the higher the beta-carotene content.

### 3.2. Body Weight and In Vivo Results

As shown in Figure 1, a diet of corn flour cookies resulted in reduced blood glucose levels in diabetic rats after four weeks. In contrast, no decrease in blood sugar levels was observed when diabetic rats were fed cookies made from wheat flour. As shown in Table 1, the dietary fibre content of corn cookies was higher than that of wheat cookies, which may have contributed to the observed decrease in blood sugar levels.

Dietary fibre is made up of the plant polysaccharide lignin, which is resistant to hydrolysis by human digestive enzymes [9]. Fibre, particularly water-soluble fibre, becomes more viscous in food, slowing the process of absorption of nutrients such as glucose. As a result, fibre consumption has a beneficial effect on blood glucose levels in people with diabetes [47]. Ahn et al. [42] proposed that fibre reduces the activity of digestive enzymes and the level of food penetration, supported by their finding that consumption of rice analogues with a fibre content of up to 22.1% significantly lowered blood sugar levels in diabetic rats (Δ 175 mg/dL). Furthermore, the addition of VCO to corn cookies has been shown to aid in the reduction in blood sugar levels in mice, while, similarly, [48] stated that the feeding of VCO lowered blood sugar levels in rats.

Hyperglycaemia causes an increase in free radicals in cells, which can be toxic in excess, encouraging oxidative stress and the formation of reactive oxygen species (ROS) as well as reactive nitrogen species (RNS) [8]. Streptozotocin, a diabetogenic, can produce reactive oxygen, which, when induced in model animals, can result in increased levels of ROS. This oxidative stress exacerbates the progression and complications of diabetes [49]. As levels of free radicals increase, so does peroxidation of cell membrane lipids and production of MDA, one of the final products of peroxidation.

Figure 2 shows that MDA levels were lower in diabetic rats that were on a diet of corn cookies rather than wheat flour cookies. The lowest MDA levels (apart from the negative control) were found in rats that were fed corn cookies containing crystalline coconut sugar and VCO (C4); the levels of MDA in these rats were almost identical to those observed in normal (non-diabetic) rats on a standard diet (negative controls). Levels of MDA are thought to be related to the antioxidant properties of beta-carotene. Importantly, C4 corn cookies made with crystalline coconut sugar and VCO were found to have the highest beta-carotene content (1877.9 ppm, see Table 1). As an antioxidant, beta-carotene can protect beta-pancreatic cells from the cytotoxicity caused by oxidative stress in DM [50]. According to Furusho et al. [51], supplementation with beta-carotene at 20 mg/kg body weight significantly reduced the levels of ROS and increased the levels of antioxidant enzymes in diabetic rats. The antioxidant activity of beta-carotene has been shown to occur indirectly through the maintenance of cell membrane integrity against free radical attacks [52].

Apart from the beta-carotene supplied by the corn flour, the low MDA content in C1 cookies was also due to VCO, which contains polyphenol components that can reduce lipid oxidation and levels of LDL. According to Amin et al. [22], VCO lowers the levels of total cholesterol, triglycerides, phospholipids, LDL, and very-low-density lipoprotein (VLDL) cholesterol while increasing levels of high-density lipoprotein (HDL) cholesterol in serum and tissues.

According to Rukmini et al. [53], more than 90% of the fatty acids in VCO consist of saturated fatty acids, especially lauric acid (12:0) in the amount of 50.5%, followed by miristic (14:0) by 17%, palmitic (16:0) by 8.5%, caprilic (8:0) by 7%, and capric (10:0) by 6.5%. Saturated fatty acids in VCO are easily metabolized by the body and the polyphenol fraction in VCO can significantly reduce lipid and LDL oxidation, thereby helping to reduce malondialdehyde, while 95% of the fatty acids in margarine consist of saturated fatty acids, either monounsaturated or polyunsaturated fatty acid [54].

As described above, MDA is a biomarker for oxidative stress that is formed as a result of the reaction between free radicals and the unsaturated fatty acids that make up cell membranes [55]. A state of oxidative stress is one of the causes of diabetes, as it can induce insulin resistance in peripheral tissues and impair insulin secretion from pancreatic beta cells [56]. Oxidative stress also leads to cell membrane lipid peroxidation. Erythrocyte haemolysis is aided by lipid peroxidation in cellular membranes, producing free haemoglobin and causing cell haemoglobin levels to fall [57].

Non-diabetic rats on the standard diet had normal haemoglobin (Hb) levels of 14.73 g/dL, whereas diabetic rats on the same diet had Hb levels of 10–11 g/dL, as shown in Figure 3. However, four weeks on a diet of corn flour cookies led to increased haemoglobin levels in the diabetic rats. The increased haemoglobin levels were comparable to normal adult women’s haemoglobin levels of 12–14 g/dL and normal adult men’s haemoglobin levels of 14–16 g/dL [31]. In contrast, haemoglobin levels decreased in the control group of diabetic rats that were fed wheat flour cookies.

The greatest increase in haemoglobin was observed in diabetic rats that were fed corn cookies sweetened with crystalline coconut sugar and VCO emulsion (Δ = 3.86 g/dL). This was likely due to the high iron content of the C4 cookies (4.7%). Iron is absorbed in the duodenum and upper jejunum via a complex process [58]: Fe^3+^ first dissolves in gastric acid in the stomach, becoming bound by gastroferrin and reduced to Fe^2+^; in the intestine, Fe^2+^ is oxidised to Fe^3+^ and binds to apoferritin, which is then converted to ferritin, releasing Fe^2+^ into the blood plasma; finally, Fe^2+^ is oxidised to Fe^3+^ in the plasma and binds to transferrin, which transports Fe^2+^ into the bone marrow, where it combines with haemoglobin. Our findings are consistent with Cabarello et al. [43], who stated that the feeding of foods high in iron, such as bay leaves, can increase haemoglobin, albumin and iron levels.

As changes in food and nutritional status have a strong influence on body weight, this is one of the anthropometric measurements that is frequently used for assessing nutritional status [28]. Insulin deficiency in diabetics is known to disrupt protein and fat metabolism, resulting in weight loss ([59].

As shown in Figure 4, normal rats on the standard diet experienced a significant increase in body weight, whereas diabetic rats on the same diet experienced significant weight loss. In contrast, the body mass index of diabetic rats that were fed corn cookies increased almost as much as that of normal rats fed the standard diet. The greatest increase in body weight (Δ 20.60 g) resulted from a diet of corn cookies sweetened with crystalline coconut sugar and margarine emulsion (C2).

These results suggest that consumption of cookies made from corn flour may be able to have a positive impact on diabetes in humans, as patients with diabetes typically lose a lot of weight. This is due to the body’s inability to provide glucose for metabolisation into energy, which leads to the burning of fat stores instead [60]. Furthermore, the inability of tissues to utilise blood glucose causes the liver to utilise more fatty acids and protein as an energy source [28]. These findings are supported by the results of Vedasree et al. [61], who found that feeding *Cyanotis tuberosa* (Roxb.) Schult as a supplement increased the body weight of diabetic rats.

Diabetes mellitus is a chronic disease characterised by high blood glucose levels. This is a result of impaired glucose homeostasis regulation due to the inability of the pancreas to produce insulin, a hormone that regulates blood glucose levels. According to Ayala et al. [62], subjects with uncontrolled type II diabetes had significantly higher levels of MDA, SOD and KGD compared with negative controls.

## 4. Conclusions

According to the findings of this study, placing diabetic rats on a diet of corn flour cookies for four weeks reduced blood sugar to levels comparable to those observed in normal rats. Further positive responses (i.e., low MDA levels, increased haemoglobin levels and body weight approaching normal) supported this result. The most effective corn flour cookies contained crystalline coconut sugar and VCO, although cookies made with granulated sugar and margarine had almost the same properties. Overall, diabetic rats were in the same physiological condition as normal rats after four weeks of being fed corn cookies containing crystalline coconut sugar and VCO. These findings suggest that such cookies are gluten-free functional foods suitable for diabetics.

## Figures and Tables

**Figure 1 foods-11-01819-f001:**
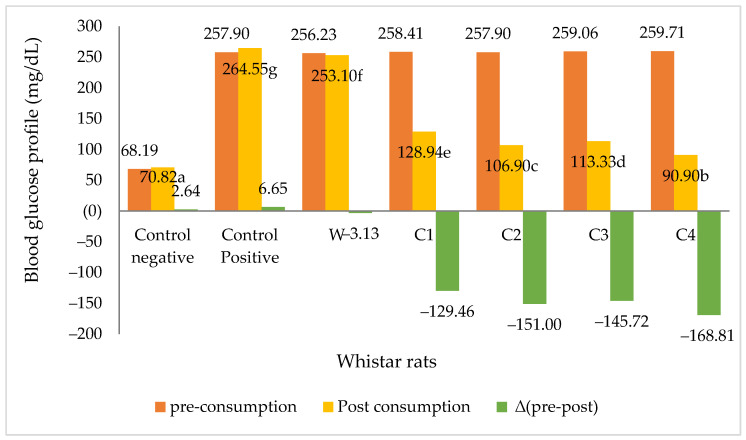
Blood glucose levels in diabetic Wistar rats after four weeks of feeding. Note: Negative control = non-diabetic rats; positive control = diabetic rats with normal diets; W = diabetic rats on wheat cookies diet; C1 = diabetic rats on C1 diet (corn cookies containing granulated sugar and margarine); C2 = diabetic rats on C2 diet (corn cookies containing crystalline coconut sugar and margarine); C3 = diabetic rats on C3 diet (corn cookies containing granulated sugar and VCO); C4 = diabetic rats on C4 diet (corn cookies containing crystalline coconut sugar and VCO). Different letters behind the numbers indicate a significant difference at the 5% level.

**Figure 2 foods-11-01819-f002:**
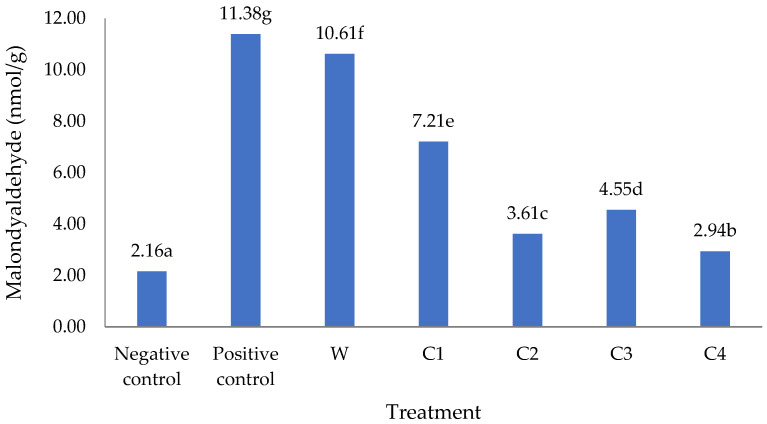
Malondialdehyde levels in diabetic Wistar rats after four weeks of feeding. Note: Negative control = non-diabetic rats; positive control = diabetic rats with normal diets; W = diabetic rats on wheat cookies diet; C1 = diabetic rats on C1 diet (corn cookies containing granulated sugar and margarine); C2 = diabetic rats on C2 diet (corn cookies containing crystalline coconut sugar and margarine); C3 = diabetic rats on C3 diet (corn cookies containing granulated sugar and VCO); C4 = diabetic rats on C4 diet (corn cookies containing crystalline coconut sugar and VCO). Different letters behind the numbers indicate a significant difference at the 5% level.

**Figure 3 foods-11-01819-f003:**
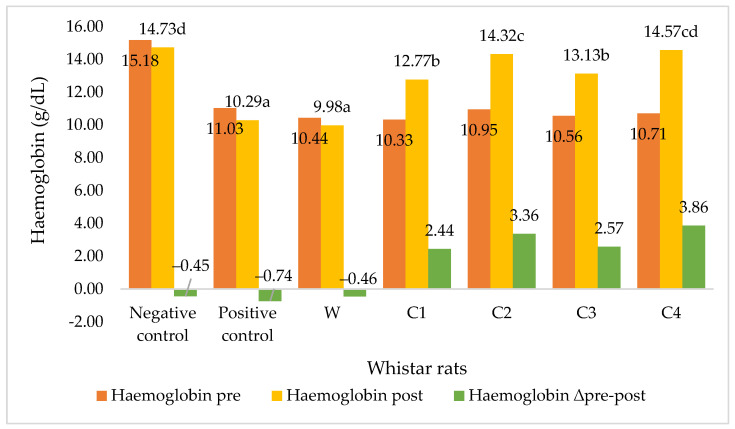
Haemoglobin levels in diabetic Wistar rats pre and post of feeding. Note: Negative control = non-diabetic rats; positive control = diabetic rats with normal diets; W = diabetic rats on wheat cookies diet; C1 = diabetic rats on C1 diet (corn cookies containing granulated sugar and margarine); C2 = diabetic rats on C2 diet (corn cookies containing crystalline coconut sugar and margarine); C3 = diabetic rats on C3 diet (corn cookies containing granulated sugar and VCO); C4 = diabetic rats on C4 diet (corn cookies containing crystalline coconut sugar and VCO). Different letters behind the number indicate a significant difference at the 5% level.

**Figure 4 foods-11-01819-f004:**
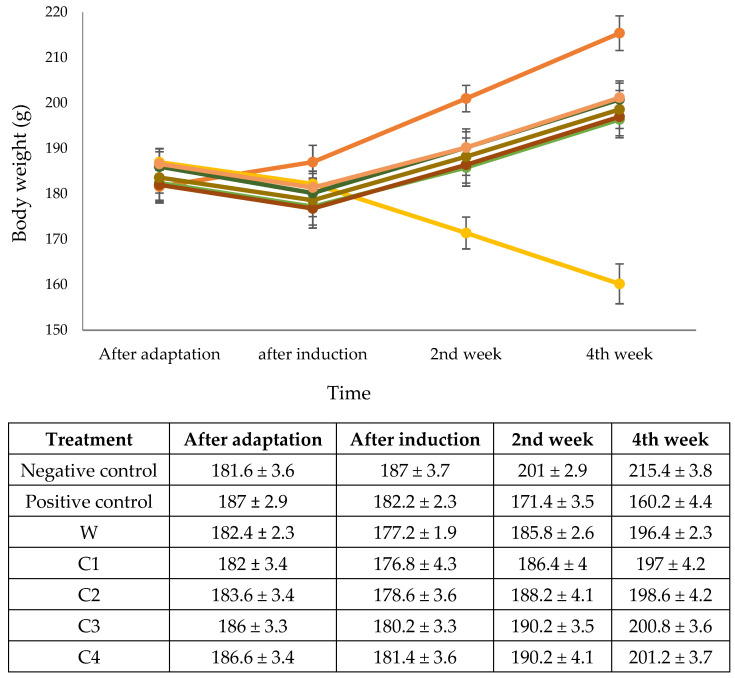
Weight body in diabetic Wistar rats pre and post of feeding. Note: negative control = non-diabetic rats; positive control = diabetic rats with normal diets; W = diabetic rats on wheat cookies diet; C1 = diabetic rats on C1 diet (corn cookies containing granulated sugar and margarine); C2 = diabetic rats on C2 diet (corn cookies containing crystalline coconut sugar and margarine); C3 = diabetic rats on C3 diet (corn cookies containing granulated sugar and VCO); C4 = diabetic rats on C4 diet (corn cookies containing crystalline coconut sugar and VCO). Different letters indicate a significant difference at the 5% level.

**Table 1 foods-11-01819-t001:** Nutritional values of corn cookies.

Parameter	W	C1	C2	C3	C4
Moisture content (%)	3.7 ^c^ ± 0.04	3.9 ^d^ ± 0.06	4.4 ^e^ ± 0.02	3.4 ^a^ ± 0.01	3.6 ^b^ ± 0.01
Ash (%)	1.3 ^a^ ± 0.04	1.8 ^b^ ± 0.02	1.9 ^b^ ± 0.07	1.5 ^a^ ± 0.2	1.8 ^b^ ± 0.03
Fat (%)	12.7 ^a^ ± 1.04	13.3 ^a^ ± 0.76	15.8 ^b^ ± 1.26	17.3 ^b^ ± 1.26	17.8 ^b^ ± 1.04
Protein (%)	29.2 ^d^ ± 1.29	23.7 ^c^ ± 0.37	19.8 ^a^ ± 0.77	21.7 ^b^ ± 0.49	19.3 _a_ ± 0.35
Carbohydrate (%)	56.7 ^a^ ± 2.23	60.9 ^b^ ± 0.87	62.3 ^b^ ± 1.89	59.4 ^a^ ± 1.09	60.9 ^b^ ± 1.11
Sugar (%)	85.4 ^e^ ± 0.68	80.4 ^d^ ± 1.11	72.4 ^c^ ± 0.26	69.2 ^b^ ± 0.60	63.9 ^a^ ± 0.75
Dietary fibre (%)	2.0 ^a^ ± 0.07	4.2 ^b^ ± 0.12	5.14 ^c^ ± 0.10	6.2 ^d^ ± 0.19	6.1 ^d^ ± 0.29
Beta-carotene (ppm)	782.3 ^a^ ± 10.72	1250.4 ^b^ ± 16.66	1591.7 ^c^ ± 16.69	1782.7 ^d^ ± 16.67	1877.9 ^e^ ± 16.55
Fe (mg/100 g)	2.5 ^a^ ± 0.04	3.7 ^b^ ± 0.04	3.9 ^c^ ± 0.03	4.5 ^d^ ± 0.03	4.7 ^e^ ± 0.03

Notes: numbers followed by the same letter within the same row indicate no significant difference at 5%; W = control wheat flour cookies made with sugar and margarine; C1 = corn cookies made with granulated sugar and margarine; C2 = corn cookies made with crystalline coconut sugar and margarine; C3 = corn cookies made with granulated sugar and VCO; C4 = corn cookies made with crystalline coconut sugar and VCO.

## Data Availability

The data presented in this study are available on request from the corresponding author.

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
