# Peer review of "Blood Sugar, Haemoglobin and Malondialdehyde Levels in Diabetic White Rats Fed a Diet of Corn Flour Cookies"

_foods, 2022, doi:10.3390/foods11121819_

Round 1
Reviewer 1 Report
The article entitled “Blood Sugar, Haemoglobin and Malondialdehyde Levels in Diabetic White Rats Fed a Diet of Corn Flour Cookies” authored by Nur Aini et al deals with the analysis of the chemical composition of corn cookies containing different types of sugar and fat, and the determination of their effect on physiological parameters in diabetic rats.
The scope of the study is interesting, however, I do have some concerns and suggestions to improve the manuscript. I hope you find these comments helpful to your manuscript and I look forward to reviewing your reply.
1. Page 2, line 45 and 50. Authors state that corn flour has low GI and they hypothesize that cookies made with corn flour will have low GI. The authors should give the glycemic index of corn flour explain why such a flour as corn flour has low GI by citing relevant references.
2. Page 2, lines 58-60. The authors state that sugar content of crystalline coconut sugar is lower than that of granulated sugar, ideal for diabetics and capable of lowering unsaturated fat levels in the body. The authors should provide relevance references for that states
3. Page 3 lines 113-114. The authors should explain why they did not record the food consumed by the experimental animals, since they are interested to record their body weight.
4. Page 4, Table 1 and page 5 lines 234-235. The authors have planned a protocol where the food received by each group of experimental animals differs in the sugar content. The authors should answer clearly if that difference does not influence the final results for blood glucose levels and Malondialdehyde levels (figures 1 and 2)
5. Figure 4. Authors should provide standard deviation for the weight measurements of experimental animals
Minor comments
1. Page 3, line 121. Authros should give ´g instead of rpm for blood sample centrifugation
2. Page 4, Table 1-first column. Authors should give unit of measurement for Fe
Author Response
Thank you for taking the time to review and provide suggestions for my manuscript. I have improved the manuscript according to your suggestion, with the comments on the attachment

Reviewer 2 Report
General comments: The manuscript is very interesting and fits in the scope of the Foods journal. The topic of the study concerns the consumption of sweets by diabetes, whose diets are restrictive and generally devoid of sweet snacks due to GI calculations. Offering an alternative to classic sweets in the form of corn cakes may be useful and help people with diabetes to function properly. The Authors suggest using coconut oil, rich in medium-chain fatty acids, instead of margarine, which will reduce the supply of saturated fatty acids. Furthermore, the authors highlighted that medium-chain fatty acids gradually regenerate pancreatic beta cells, stimulating insulin production and improving insulin sensitivity. Instead of sucrose, they suggest introducing coconut sugar.
1. Abstract: written appropriately
2. Key words: correct
3. Introduction: presents the topic in an appropriate way; the authors prove the validity of introducing coconut oil instead of margarine, coconut sugar instead of sucrose and the partial replacement of wheat flour with corn flour. However, I believe that information on the different types of diabetes should be provided, indicating which type the authors are targeting. Oxidative stress in diabetes should also be mentioned to justify the determination of MDA and Fe.
4. Purpose of the study: well formulated; the authors set two objectives for the work: (1) determine the chemical composition of corn cookies containing different types of sugar and fat and (2) examine their effect on blood sugar levels, MDA, haemoglobin, and body weight in diabetic rats
5. Materials and methods: described in great detail; however, it should be added whether the recipe for the cookies was developed by the authors; no description of the statistical analysis
6. Results and discussion: well described and discussed results.
- line 183 please change to „In addition, Dewi et al. [10] demonstrated higher…”
- line 271 proszÄ™ zamienić na „ Ahn et al. [37] proposed that fibre…”
- similar line 275, 293, 306, 339, 363, 368 and others
- references should be cited in accordance with the editorial requirements (line 156 and 158)
- fatty acid analysis is somewhat lacking in the results - did the authors perform such analyses? If not, it may be worth at least comparing the fatty acid profile of coconut oil and margarine based on the literature
7. Conclusions: correct
8. References: I believe that relevant literature was used in the study to explore the issues fully.
9. Figures and tables: correct
Author Response
Thank you for taking the time to review and provide suggestions for my manuscript. I have improved the manuscript according to your suggestion, with the comments on the attachments

Round 2
Author Response
Thank you for taking the time to review and provide suggestions for my manuscript. I have improved the manuscript according to your suggestion, with the following comments:
Comment 1: Authors should provide standard deviation values for the weight measurements of experimental animals also in the table below the graph in Figure 4
Response: I've added the standard deviation to Figure 4, but sorry I changed the way it's written, because if I wrote it using yesterday's method I couldn't edit and add the standard deviation value.
Comment 2: Authors are advised to provide the value of 210 g as 210 x g
Response: Have been done, thank you
Thank you very much
